# Correlation of Fabrication Methods and Enhanced Wear Performance in Nanoporous Anodic Aluminum Oxide with Incorporated Molybdenum Disulfide (MoS_2_) Nanomaterials

**DOI:** 10.3390/nano14050451

**Published:** 2024-02-29

**Authors:** Kendrich O’Donaghue Hatfield, Nathan Brown, Enkeleda Dervishi, Bradley Carpenter, Jordyn N. Janusz, Daniel E. Hooks

**Affiliations:** 1Finishing Manufacturing Science, Los Alamos National Laboratory, Los Alamos, NM 87545, USA; khatfield@lanl.gov (K.O.H.);; 2Center for Integrated Nanotechnologies, Los Alamos National Laboratory, Los Alamos, NM 87545, USA

**Keywords:** anodized aluminum, molybdenum disulfide, wear resistance, friction coefficient, thin films

## Abstract

Wear performance is integral to component longevity, minimizing industrial waste and excess energy costs in a wide variety of applications. Anodized aluminum oxide (AAO) has many beneficial properties leading to its wide use across industries as a surface treatment for many aluminum components, but the wear properties of the coating could be improved significantly. Here, we used an electrochemical method to incorporate molybdenum disulfide (MoS_2_), a nanomaterial used as a dry lubricant, to modify alloys of aluminum during AAO preparation. Using Raman spectroscopy and tribological scratch measurements, we thoroughly characterized the structure and wear behavior of the films. The MoS_2_ deposition procedure was optimal on aluminum 5052 anodized in higher acid concentrations, with friction coefficients at around 0.05 (~10× better than unmodified AAO). Changing anodization conditions to produce harder films with smaller pores led to worsened wear properties, likely because of lower MoS_2_ content. Studying a commercial MoS_2_/AAO film of a different Al alloy (7075) showed that a heat treatment step intended to fully convert all deposited MoS_x_ species to MoS_2_ can adversely affect wear in some alloys. While Al 6061 and 1100 produced films with worse wear performance compared to Al 5052 or 7075, our results show evidence that acid cleaning after initial anodization likely removes residual alloying elements, affecting MoS_2_ incorporation. This study demonstrates a nanomaterial modified AAO film with superior wear characteristics to unmodified AAO and relates fabrication procedure, film structure, and practical performance.

## 1. Introduction

Materials with good wear and friction characteristics are critical to reducing waste and overall process costs in many industrial sectors. Anodized aluminum oxide (AAO) has been used as a coating for aluminum extensively in industries for many years owing to its ready availability, strength, and corrosion resistance [1]. Previous work in our laboratory showed how differing anodization treatments affect the nanoporosity, wear rate, and friction coefficients of the resulting films [2]. Though AAO has significant history, its fabrication process and applications remain of current interest [3]. Researchers have refined AAO nanopore engineering and modification for various applications like sensors [4], catalysis [5,6], and wear [7,8]. MoS_2_, existing naturally in the form of molybdenite, has been used as a lubricant for various materials for decades [9,10]. Processes to include MoS_2_ in mixtures and novel coatings have been evaluated [11,12,13,14,15]. Additional studies have shown the use of nanoparticle forms of MoS_2_ and other materials for property modification in general and with specific application to the functional modification of AAO [3,16,17]. Researchers have inserted MoS_2_ into AAO pores via direct electrophoretic deposition [18] and used electrochemically induced insertion for soluble MoS_x_ species that are subsequently converted to MoS_2_ by thermal or plasma-induced decomposition in an O_2_-free atmosphere [19,20,21,22]. A better understanding of how materials insert into AAO nanopores and the resulting material distribution will help the rational design of future AAO-based functional materials.

A patent by Saruwatari et al. outlines a procedure for the in situ incorporation of MoS_2_ into porous AAO films using a tetrathiomolybdate precursor [23]. Briefly, aluminum is anodized in an acid bath to form porous AAO before further anodization in a near neutral (pH 6–9) solution containing ammonium tetrathiomolybdate, (NH_4_)_2_MoS_4_. The MoS_4_^2−^ reacts with protons evolved at the aluminum surface during anodization (Equation (1)) and forms a mix of a solid MoS_2_/MoS_3_ precipitate in the AAO pores (Equation (2)), which can be fully thermally converted to MoS_2_.
(1)$$2Al+3H_{2}O\to {Al}_{2}O_{3}+6H^{+}+6e^{-}$$
(2)$$MoS_{4}^{2-}+2H^{+}\to MoS_{3}+H_{2}S$$

Figure 1 depicts the fabrication process flow. This method is industrially scalable and applicable to parts with shapes that might make the physical burnishing application ineffective. Other studies have used this method to incorporate MoS_2_ or MoS_2_ precursors (i.e., MoS_3_) into porous alumina films to improve friction and wear characteristics to varying degrees of success [24,25,26]. However, the literature currently lacks a methodical analysis of how fabrication parameters, film structure, and tribological characteristics relate. Here, we aim to clarify how different Al alloys and treatment parameters affect MoS_2_/AAO film structure for practical wear reduction. Few studies have characterized the distribution of MoS_x_ as a function of film depth, and none have correlated this distribution directly with wear properties. We use Raman microscopy to characterize MoS_2_/AAO film surfaces and cross sections, simultaneously determining the MoS_x_ species and location within the film.

Film fabrication variables include anodization conditions, the nitric acid cleaning step before MoS_x_ deposition, the thermal conversion of MoS_3_ to MoS_2_, and the aluminum alloy type. Anodization time primarily controls AAO thickness [2,24], while temperature and acid concentration affect AAO pore structure [2,27]. HNO_3_ cleaning after initial anodization may remove residual acid sulfates [27] and/or alloying elements from the porous AAO. The elemental content of different Al alloys may affect the resulting AAO film structure and wear performance. We investigate these parameters with Raman spectroscopy and tribological scratch measurements to assess film structure and wear performance. In general, we found that Al 5052 modified with MoS_2_ delivered the best wear properties, with low friction coefficients and the highest film breakthrough times, and our parametric study gives insight into the relation between preparation conditions, film structure, and wear resistance.

## 2. Materials and Methods

### 2.1. Materials and Chemicals

Ammonium tetrathiomolybdate ((NH_4_)_2_MoS_4_, ATTM, 99.95%, Acros Organics, Geel, Belgium), sodium hydroxide (NaOH, VWR), sulfuric acid (H_2_SO_4_, 95–98%, J.T. Baker, Sanford, ME, USA), and nitric acid (HNO_3_, 65%, Millipore-Sigma Emplura, Burlington, MA, USA) were used as received. Deionized water was obtained from a Millipore nanopure filter and used for all experiments. Aluminum 5052, 6061, 7075, and 1100 coupons were used as working electrodes, and aluminum 5052 was used as a counter electrode for both anodization and MoS_3_ insertion in a two-electrode setup. A Keysight N5770A (Keysight Technologies, Santa Rosa, CA, USA) power supply was used for initial coupon anodization to form AAO, and a BK Precision PVS10005 (Los Angeles, CA, USA) was used for MoS_x_ deposition, while the potential was monitored with a Keyence NR-X100W (Osaka, Japan) data collection unit with a Keyence NR-HV04 (Osaka, Japan) high voltage measurement attachment. Cross-sections of the Al coupons were mounted in epoxy, which, after curing, were polished, finishing with a 1 µm colloidal diamond suspension.

### 2.2. Anodization Procedure

Al coupons were first cleaned in 1.5 M NaOH heated to 70 °C for about 1 min to remove surface oxides. After rinsing in water, the coupons were then soaked in 32.5 *v*/*v*% HNO_3_ for 1 min for cleaning and neutralization of residual NaOH. Aluminum coupons were then anodized for varying amounts of time at 17 V in 10 *v*/*v*% H_2_SO_4_ at 12–15 °C (high-acid anodization) or 9.3 mA/cm^2^ in 5 *v*/*v*% H_2_SO_4_ at 2–5 °C (low-acid anodization). Acid pretreatment, performed on some coupons, consisted of a 5-min soak in 6.5 *v*/*v*% HNO_3_ after initial anodization but before MoS_x_ deposition. The Al was then rinsed with water and re-anodized in 15 mM ATTM at 0.8 mA/cm^2^ to deposit MoS_3_ according to equation 2. To convert MoS_3_ to MoS_2_, the anodized coupons were placed in a furnace under N_2_ flow (500 SCCM) at 450 °C for 5 h. In general, all samples were prepared in triplicate to test result repeatability and consistency.

### 2.3. Sample Analysis

Reciprocating scratch tests were performed with an RTEC MFT-5000 (Santa Ana, CA, USA) tribometer with a 3 mm steel ball, 15 N load force, scratch speed of 6 mm/s, and a total distance of 2 m over 500 cycles (4 mm scratches). This instrument monitors force and resistance to displacement, delivering friction coefficient vs. time. A Keyence VK-X3000 (Osaka, Japan) optical profilometer was used to determine scratch volumes for film breakthrough rates. A Horiba Xplora Plus Raman microscope with an Olympus MPlan N 100× (Kyoto, Japan) (NA: 0.90, 3.5 µm spot diameter) or 10× objective (NA: 0.25, 25 µm spot diameter) and a 532 nm laser was used to characterize sample surfaces and cross-sections to confirm and locate MoS_x_ throughout the film depth. Raman spectra had varying levels of background, likely due to the reflectivity of the polished Al/AAO surfaces, so a polynomial background was fitted and subtracted in the Horiba software (Labspec version 6) for most spectra for cross-sectional plotting. An Apreo 1 or 2 (Thermo Fisher Scientific, Waltham, MA, USA) with an EDAX (Mahwah, NJ, USA) or Oxford EDS (Oxfordshire, UK) attachment was used to collect SEM images and EDS spectra. X-ray fluorescence was performed with a Thermo Scientific Niton XL3t GOLDD+ (Waltham, MA, USA) handheld spectrometer. A Keyence EA-300 (Osaka, Japan) was used to perform laser-induced breakdown spectroscopy (LIBS) with a 355 nm laser with an energy output of 100 µJ/pulse. The laser spot size was ~5 µm in diameter.

## 3. Results and Discussion

### 3.1. High-Acid Anodized Aluminum 5052

We prepared high-acid anodized MoS_2_/AAO samples with initial anodization times of 20 and 120 min to form both thin and thick AAO films, respectively (Table 1). For both anodization times, we prepared samples with and without a 5-min soak in 6.5 v/v% HNO_3_ just before MoS_x_ deposition to study the effect of an acid pretreatment, as mentioned by Saruwatari [23]. Possible effects of this post-anodization treatment include removal of residual acid sulfates [27] or alloying elements. SEM images in Appendix A show the morphology of films before (a) and after (b) MoS_2_ modification (after heat treatment). The primary morphology change we observed after MoS_2_ modification is the appearance of small globules (likely MoS_2_) across the surface.

Figure 2 shows the voltage-time (V-t) curves for the MoS_x_ deposition step for 120-min anodized MoS_2_/AAO (Appendix A shows the V-t curves for 20-min anodized MoS_2_/AAO). We stopped MoS_x_ deposition when the V-t curve (1) reached a set voltage limit of 180 V (e.g., Figure 2a), (2) showed rapid voltage fluctuations accompanied by a decrease in slope, signaling dielectric breakdown of the film (e.g., Appendix A, HNO_3_ presoak sample 2) [24,25], or (3) reached a plateau/inflection point not quickly followed by an increase in voltage (e.g., Figure 2b, sample 3). Depositions for 120-min anodized AAO films generally lasted longer than for 20-min anodized AAO films, and we observed a voltage plateau from 80–120 V for most of the 120-min AAO films before they continued to 180 V. Acid pretreatment generally increased deposition time for 120-min anodized samples while having no consistent effect on 20-min anodized samples. We did not observe that these V-t curve differences correlated to wear performance of the coatings but have included them to show process conditions.

Figure 3 shows example reciprocating scratch tests for each sample treatment with asterisks denoting film breakthrough (Appendix A shows all scratch tests for each sample). Wear behavior for most sample treatments varied significantly and generally did not appear to correlate with deposition time or V-t curve shape as depicted in Figure 2. Such variability is apparent in many of the results in Figure 3, wherein many of the coatings failed via breakthrough at short (less than 5 min) scratch times. Table 2 shows summarized reciprocating scratch results for 20-min and 120-min anodized MoS_2_/AAO samples. We define MoS_2_/AAO film breakthrough as when the coefficient of friction (COF) reached that of the as-received Al (0.73 for Al 5052, Appendix A and Table 3). We determined breakthrough rates by dividing film thickness by breakthrough time; breakthrough scratch percent indicates how many scratches showed film breakthrough. Breakthrough rates did not lead to clear correlations between treatment procedures because of high variance (standard deviations of 100% or greater), but breakthrough scratch percent indicated that certain treatments more consistently resulted in films with high wear resistance. The 120-min anodized MoS_2_/AAO films had fewer breakthroughs than 20-min anodized films. Acid pretreatment led to 20% less film breakthrough on the 20-min anodized samples but 20% more breakthrough on the 120-min anodized samples. Figure 4 highlights the superior wear properties of one of the better performing 20-min acid pretreated MoS_2_/AAO samples over Al 5052 as received or 20-min anodized AAO without MoS_2_ modification. Notably, the friction coefficient of scratches that did not break through was not significantly affected by treatment procedure (generally ~0.05), implying that whenever MoS_2_ presence is sufficient to stave off wear, it provides excellent lubrication properties. Overall, the non-pretreated 120-min anodized samples showed the least film breakthrough, though our results indicate that MoS_2_ modification can lead to highly varying tribological results. Thus, industrial manufacturing methods need considerable development to produce films with consistent performance.

### 3.2. Raman Spectroscopy and Cross-Sectional Analysis

To study the distribution of MoS_2_ in MoS_2_/AAO films, we performed Raman spectroscopy on MoS_2_/AAO surfaces and cross-sections. Figure 5 shows the spectrum measured from the surface of a 20-min anodized MoS_x_/AAO sample before and after heat treatment (Appendix A shows surface spectra obtained for all treatments). Before heat treatment, we observe peaks for Mo-Mo stretching (226 cm^−1^), Mo-S stretching (286 cm^−1^, 324 cm^−1^, and 355 cm^−1^), MoS_2_ E12g vibration (380 cm^−1^), MoS_2_ A1g vibration (402 cm^−1^), MoS_2_ longitudinal acoustic mode (449 cm^−1^), terminal S-S stretching (520 cm^−1^), and bridging S-S stretching (550 cm^−1^) [28,29]. After heat treatment, many of these peaks vanished, leaving peaks we assign to Mo-Mo stretching, MoS_2_ E12g, and A1g vibrations, as expected for MoS_2_. We analyzed film cross-sections with Raman spectroscopy to assess the presence and relative amount of MoS_2_ throughout the MoS_2_/AAO film depth. Figure 6a shows an optical image of a cross-section with markings showing the points where spectra were collected, and Figure 6b–e shows Raman spectra as a function of film depth for both 20-min and 120-min samples (see Appendix A for all Raman depth profiles plotted next to reciprocating scratch data for high-acid anodized Al 5052). We also performed LIBS on several sample cross-sections, plotted in Appendix A, which validate the MoS_2_ presence and concentration trends of Raman via detection of elemental Mo, albeit with lower spatial resolution and the inability to distinguish different forms of Mo.

Generally, samples with more prominent MoS_2_ peaks (380 cm^−1^ and 402 cm^−1^) throughout the depth of the film had better wear characteristics (i.e., no breakthrough or longer breakthrough times). For instance, the 20-min anodized, acid pretreated sample #3 had clear, strong Raman signals evenly through the film, and only 50% of the scratches broke through the MoS_2_/AAO. Many of the 20-min anodized samples, both with and without acid pretreatment, had the strongest Raman signals near the base of the MoS_2_/AAO film, supporting the previously proposed deposition mechanism of filling from the bottom of the pores to the surface [25]. In contrast, the 120-min anodized samples showed more varied Raman strength profiles, with several samples having consistent signal strength throughout the film and a few samples with stronger signal near the surface. We note that nearly all 120-min anodized samples that showed consistent MoS_2_ signal throughout the films resulted in little to no breakthrough, while most 120-min anodized samples that showed a decrease in MoS_2_ signal towards the surface had sample breakthrough. These results indicate that MoS_2_/AAO wear properties correlate with overall film structure, with films containing higher and even MoS_2_ content generally yielding the best results.

### 3.3. Heat Treatment Effects

While the original patent detailing MoS_2_ modification of AAO included a heat treatment step in a vacuum or an inert gas to fully convert the MoS_x_ precursor to MoS_2_ [23], several tribology studies of MoS_x_/AAO films using this method did not include heat treatments [24,25]. To study structural and wear trends with heat treatment, we conducted cross-sectional Raman and tribological tests on several 120-min high-acid anodized Al 5052 MoS_2_/AAO samples before and after heat treatment (example results shown in Figure 7, all results shown in Appendix A). The Raman signatures for the non-pretreated samples show significant amounts of MoS_3_/MoS_2_ mixed material throughout the film before heat treatment, either stronger at the surface (samples #1 and #3) or at the base (sample #2). Interestingly, heat treatment of these samples did not lead to significant MoS_2_ peak detection in the cross-sections, and the scratch tests generally showed faster breakthrough than before heat treatment. Heat treatment on acid presoaked samples showed clear transformation from MoS_3_/MoS_2_ mixed material to MoS_2_, but still led to generally faster film breakthrough. Breakthrough rates and breakthrough scratch percent are shown in Figure 8. Acid presoaking appears to have only minor (if any) effects on tribological properties, while heat treatment increases variability in the results, adversely affecting wear resistance overall. Scratch tests on AAO films without MoS_2_ show similar increases in variability and worsening wear characteristics as shown in Table 2 and Appendix A, suggesting that heating with this procedure affects the AAO itself. These results indicate that consistently performing MoS_2_/AAO films require refinement and optimization of the heat treatment process (e.g., atmosphere, gas flow rate, temperature, duration, etc.).

### 3.4. Commercial MoS_2_/AAO Film Comparison

We compared our MoS_2_/AAO films to a commercially available MoS_2_-impregnated AAO coating. Appendix A shows the EDS spectrum and x-ray fluorescence results of the unmodified aluminum center of the part; the high presence of Zn and overall composition are consistent with Al 7075 (elemental contents of all alloys studied in this paper are listed in Table 3). The Raman analysis of a cross-section revealed prominent signatures from MoS_3_, implying that MoS_x_ was not thermally converted to MoS_2_. In addition, MoS_x_ was only detectable at the base of the film (Figure 9a). Still, reciprocating scratch measurements (Figure 9b) showed relatively low initial friction coefficients (0.1–0.2) and only partial breakthrough (parts of scratches were ~20 µm deep as shown in Appendix A compared to the ~15 µm film thickness), with final friction coefficients of ~0.5. Like some of our results with Al 5052, heat treating this commercial part converted MoS_3_ to MoS_2_ (Appendix A shows cross-sectional Raman spectra) but reduced wear resistance, evident by the near instant breakthrough upon scratching. To compare our MoS_2_/AAO fabrication method, we performed 120-min high-acid anodization on Al 7075 with and without HNO_3_ presoaking followed by MoS_x_ deposition, characterizing the resulting film before and after heat treatment. We observed that HNO_3_ pretreatment had a minor effect on the V-t deposition curve, lengthening it by decreasing the slope (Appendix A). Figure 9c shows an example Raman cross-section of non-pretreated MoS_2_/AAO from Al 7075 before heat treatment, and Figure 9d shows reciprocating scratch tests (Appendix A shows all Raman cross-sections and their respective scratch tests). Again, heat treatment consistently raised friction coefficients and increased the breakthrough rate of non-pretreated Al 7075 MoS_2_/AAO. The heat treatment of unmodified Al 7075 (i.e., no MoS_2_) did not appear to affect breakthrough rate but did increase the overall friction coefficient from 0.4 to 0.7 (Appendix A), indicating worsened wear properties. We note that the sample with the most prominent Raman peaks indicates that MoS_2_ had the highest breakthrough times. The heat treatment on HNO_3_ presoaked samples had mixed effects on the wear resistance, either improving (sample #1) or having little effect on wear (samples #2 and #3). We did not observe clear MoS_2_ or MoS_3_ Raman peaks throughout most of the acid pretreated Al 7075 films. Figure 8 summarizes the tribological results of Al 7075 with and without acid pretreatment/heat treatment. Our studies on Al 7075 and the commercial MoS_2_/AAO films indicate that alloy composition can substantially affect the film structure and wear performance, implying that treatment procedures should be individually developed for a given materials application to optimize film properties.

### 3.5. Low-Acid Anodized Aluminum 5052

To study the effects of anodization conditions on wear behavior and film structure, we performed a 120-min anodization of aluminum 5052 at a constant current (9.3 mA/cm^2^, with voltages generally settling between 20−30 volts during anodization) in 5 *v*/*v*% H_2_SO_4_ at 2–5 °C to more closely reflect the hard anodization conditions used industrially to form AAO (referred to here as low-acid anodization). Lower acid concentrations and temperatures cause lower ionic solubility, leading to harder, more dense films with smaller pores [2,17], which could impact MoS_2_ modification and the resulting MoS_2_/AAO wear properties. Appendix A, which displays the V-t curves for MoS_x_ deposition on these samples, does not show a voltage plateau like 120-min high-acid anodized Al 5052. Compared to the high-acid anodized samples, the deposition times were more varied, although we observed earlier dielectric breakdown in the non-pretreated samples compared to the HNO_3_ presoaked samples. Reciprocating scratch measurements showed 100% film breakthrough (Figure 10), suggesting that the denser film may not allow as much MoS_2_ insertion as the high-acid anodized samples, which leads to inferior wear properties. Cross-sectional Raman spectroscopy confirms the lack of significant MoS_2_ content throughout the film, as shown alongside the tribological data in Appendix A.

### 3.6. Aluminum Alloy Study

To further study the effects of alloy content on film structure and wear properties, we prepared and characterized MoS_2_/AAO films from Al 6061 (primary alloying elements: Mg and Si) and Al 1100 (nearly pure aluminum) using high-acid anodization for 120 min. Table 3 shows the elemental composition of each alloy given by either the manufacturer’s certificate of analysis (Al 1100) or a listed standard [30]. Al 1100 showed a slight trend in lengthened deposition times in the V-t deposition curve with HNO_3_ presoaking, while 6061 did not show any trends (Appendix A). Since Al 1100, a nearly pure Al alloy, showed a V-t curve trend with acid presoaking, this treatment likely has effects other than alloying element removal. Example Raman cross-sectional analyses of non-pretreated MoS_2_/AAO samples are shown in Figure 11, and the tribological scratch data alongside Raman cross-sections for aluminum 6061 and 1100 are shown in Appendix A. All Al 6061, Al 1100, and nearly all Al 7075 samples showed film breakthrough, indicating inferior wear behavior to Al 5052. In general, MoS_2_/AAO films from these alloys contained the majority of MoS_2_ near the film base, tapering off to barely- or non-discernable signals near the surface. Raman cross-sections of MoS_2_/AAO samples prepared with Al 6061 showed little MoS_2_ content except for one pretreated and one non-pretreated sample. Notably, the Al 6061 sample with the highest Raman peaks (non-pretreated, sample #1, Appendix A) for MoS_2_ showed the lowest breakthrough rate (~0.5 µm/s), but still had friction coefficients of ~0.5 before film breakthrough, which is similar to unmodified AAO. The Raman analysis of Al 1100 showed results with no clear trends for MoS_2_ content between anodization conditions or pretreatment; 1–2 samples of each treatment type showed significant MoS_2_ content throughout the film. All scratches broke through, but breakthrough rates were all under 1 µm/s. Figure 10 shows the summarized wear properties of all Al alloys studied. Altogether, our results demonstrate that MoS_2_/AAO film structures and wear resistance vary widely based on initial anodization conditions, acid treatment (or lack thereof), and alloy content. Our treatment procedure led to the lowest friction coefficients and longest breakthrough times with non-pretreated 120-min high-acid anodized Al 5052, while other alloys may require procedural adjustment for wear performance optimization.

## 4. Conclusions

We prepared MoS_2_/AAO films using a variety of treatment parameters and aluminum alloys, assessing their structure and wear properties with Raman spectroscopy and reciprocating scratch measurements. The films with the best wear properties were obtained with high-acid anodized aluminum 5052, some of which maintained a friction coefficient of ~0.05 over the entire course of the scratch measurement with no film breakthrough. Acid pretreatment increased variability of the structure and behavior of the final film. Heat treatment to convert MoS_3_ to MoS_2_ also produced highly varying results, often increasing the breakthrough rate and/or friction coefficients. Heat treatment on unmodified AAO also showed worsening wear characteristics, indicating that the AAO itself may be adversely affected by the process. We used Raman spectroscopy to measure MoS_2_ content as a function of film depth and found that more MoS_2_ content, especially if it was present throughout the entire film, correlated with improved wear resistance. This cross-sectional Raman analysis method could be applied to other functional thin films to correlate structure with performance. Different initial anodization conditions, using lower acid content, higher current/voltage, and lower temperature, led to worsened wear results, likely due to the smaller pores in AAO allowing less MoS_2_ content in the film. The examination of different aluminum alloys, including a commercial MoS_2_/AAO film, showed that aluminum alloys can profoundly affect film structure and wear performance. Commercial implementations of this AAO/MoS_2_ coating are apparently not heat treated for final conversion to MoS_2_. Though the wear performance is better than that of just AAO, it is not as good as it could be with heat treatment. Presumably, this is done partially for the sake of appearance, as heat treatment converts the film from a distinct gold/brown coating to a silvery coating indistinguishable from untreated AAO. However, our results show that sometimes the full conversion to MoS_2_ through heat treatment led to worse wear performance, possibly due to the degradation of the AAO structure itself. This is consistent with many of our findings. The specific wear performance is optimized through a combination of anodization type, acid pre-treatment, and heat treatment, and the optimum process appears to be unique for each individual alloy. Though the relationships are complex and not consistent from alloy to alloy, the results here demonstrate the steps necessary to achieve the most wear-resistant AAO/MoS_2_ coating on several Al alloys. Our investigation on treatment parameters and Al alloy content in MoS_2_/AAO films gives fundamental insight on deposition mechanisms and film structure, which could guide future development of nanoporous materials for various applications.

## Figures and Tables

**Figure 1 nanomaterials-14-00451-f001:**
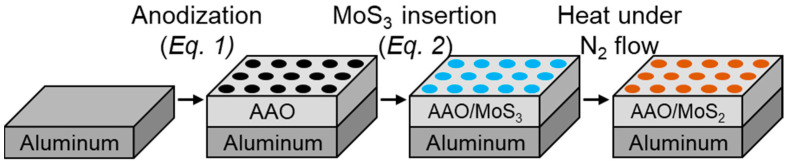
Illustration of general fabrication procedure for MoS_2_/AAO films on Al coupons.

**Figure 2 nanomaterials-14-00451-f002:**
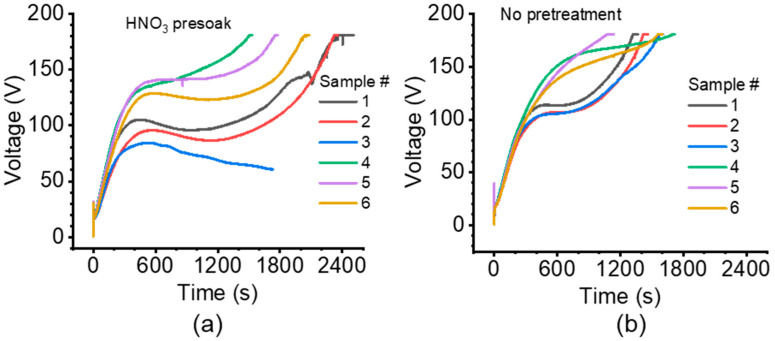
V-t curves for MoSx deposition on 120-min high-acid anodized Al 5052 with (**a**) and without (**b**) HNO_3_ pretreatment. Six samples were prepared for each treatment (sample numbers denoted in the figure).

**Figure 3 nanomaterials-14-00451-f003:**
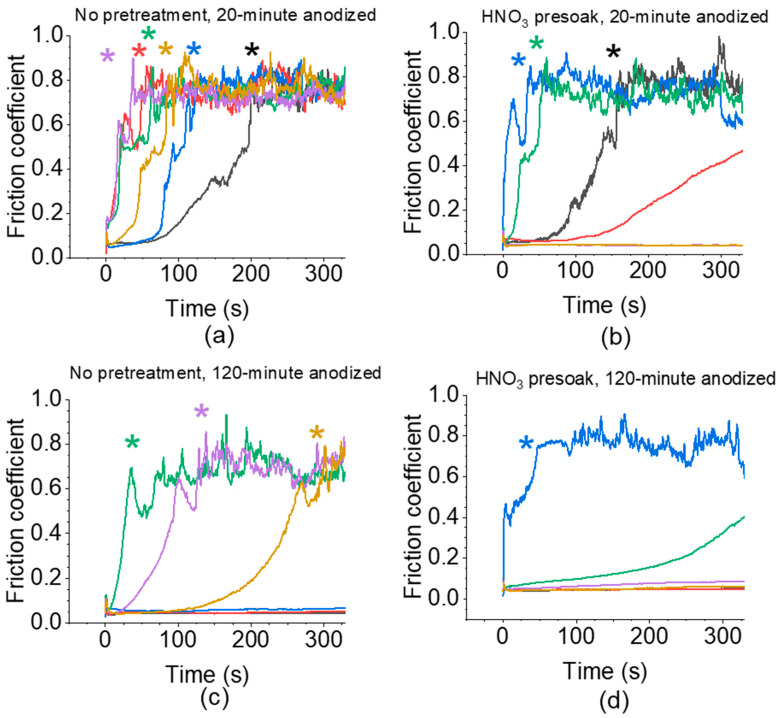
Example reciprocating scratch tests for high-acid anodized Al 5052 MoS_2_/AAO samples. Both sides of three samples were scratched for a total of six sets of measurements for each fabrication procedure (different colors represent scratches from different measurement sets). (**a,b**) 20-min anodized samples without (**a**) and with (**b**) an HNO_3_ presoak step. (**c,d**) 120-min anodized samples without (**c**) and with (**d**) an HNO_3_ presoak step. Film breakthrough is noted by asterisks on the plot.

**Figure 4 nanomaterials-14-00451-f004:**
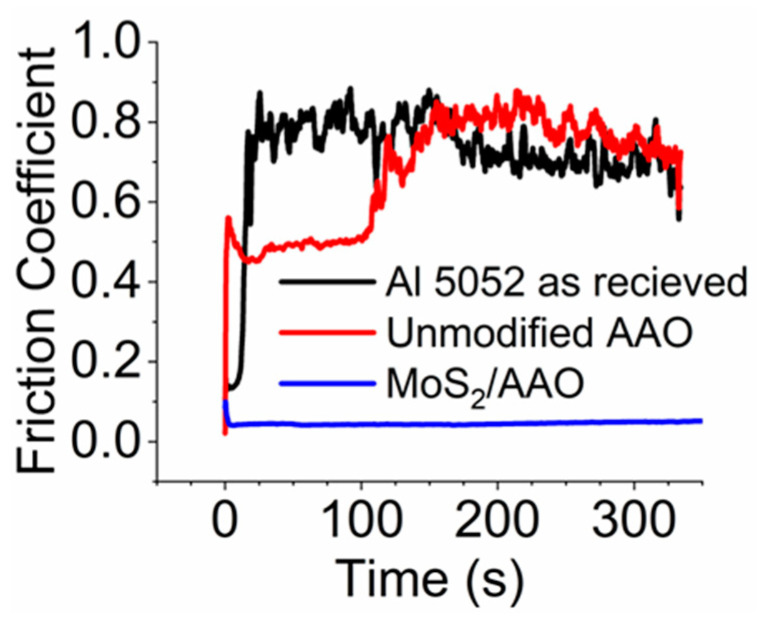
Friction coefficient comparisons obtained via reciprocating scratch measurements for Al 5052 as received, AAO with no MoS_2_ modification, and MoS_2_/AAO (with acid pretreatment). Both AAO samples were 20-min anodized.

**Figure 5 nanomaterials-14-00451-f005:**
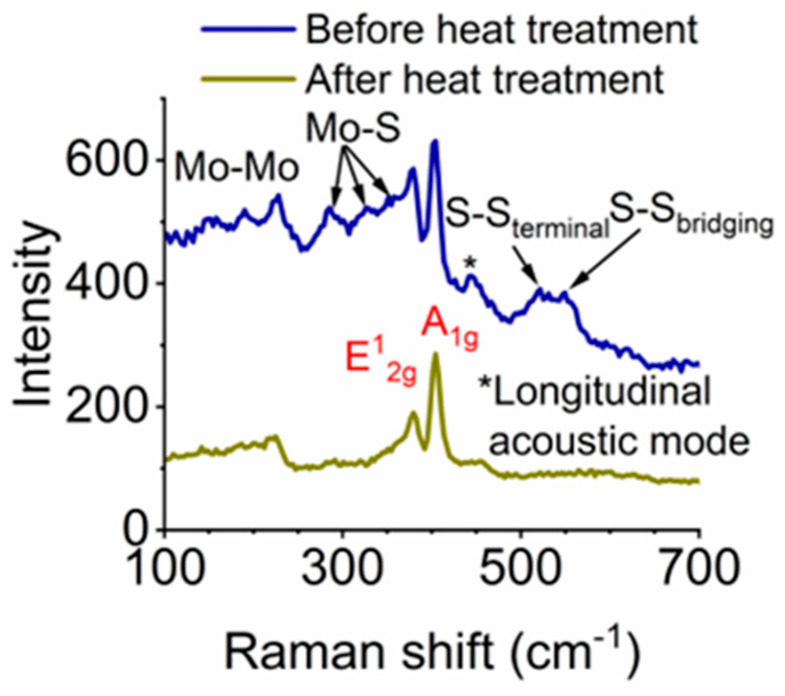
20-min anodized, acid pretreated samples before and after heat treatment at 450 °C for 5 h under N_2_ flow to convert MoS_3_ to MoS_2_.

**Figure 6 nanomaterials-14-00451-f006:**
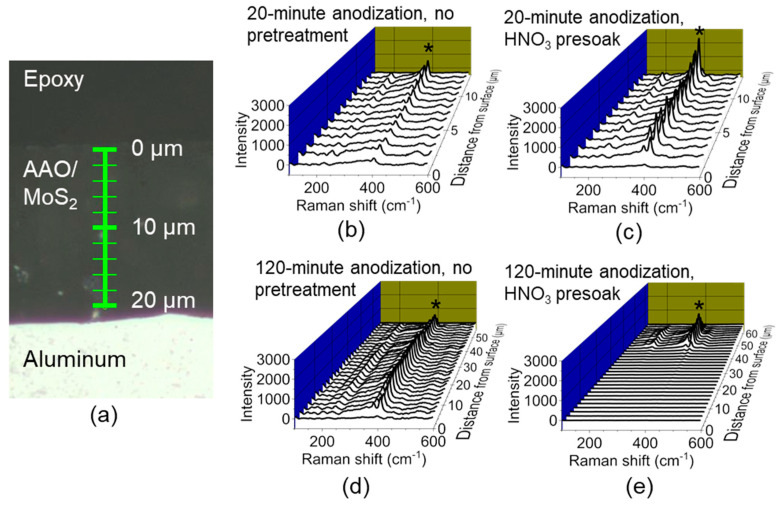
Cross-sectional Raman spectroscopic analysis of MoS_2_/AAO fabricated from high-acid anodized Al 5052. (**a**) Optical image of cross-section with waterfall plots showing the Raman spectra of MoS_2_/AAO films fabricated on Al 5052 anodized for (**b**) 20 min with no further pretreatment, (**c**) 20 min followed by acid pretreatment, (**d**) 120 min with no further pretreatment, and (**e**) 120 min followed by acid pretreatment. The A1g peak indicating MoS_2_ presence is denoted with an asterisk.

**Figure 7 nanomaterials-14-00451-f007:**
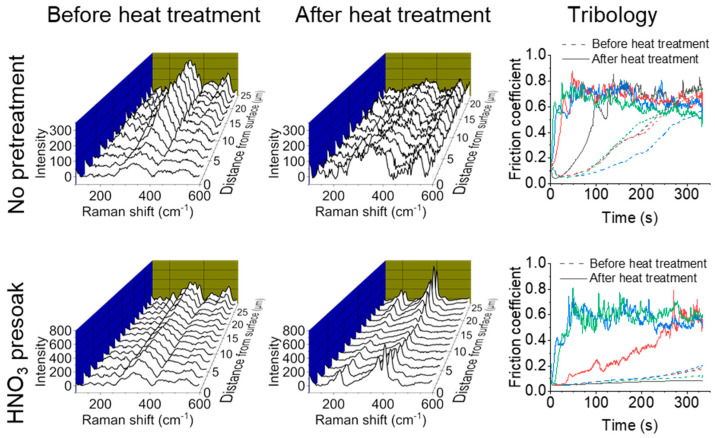
Example Raman cross-sections and tribological tests of MoS_2_/AAO fabricated from high-acid anodized Al 5052 with no pretreatment (**top**) and a HNO_3_ presoak (**bottom**) before and after heat treatment. Different colors in the scratch tests indicate different samples.

**Figure 8 nanomaterials-14-00451-f008:**
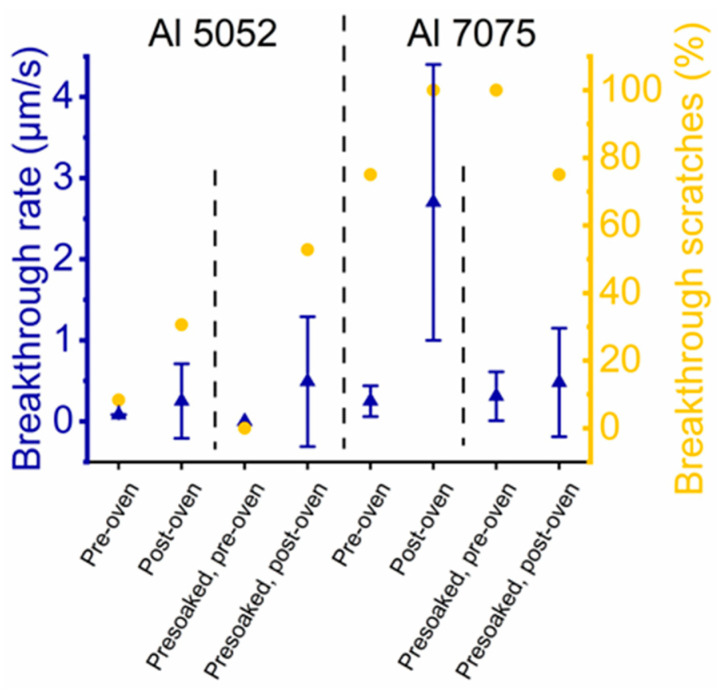
Reciprocating scratch results of MoS_2_/AAO films before and after heat treatment.

**Figure 9 nanomaterials-14-00451-f009:**
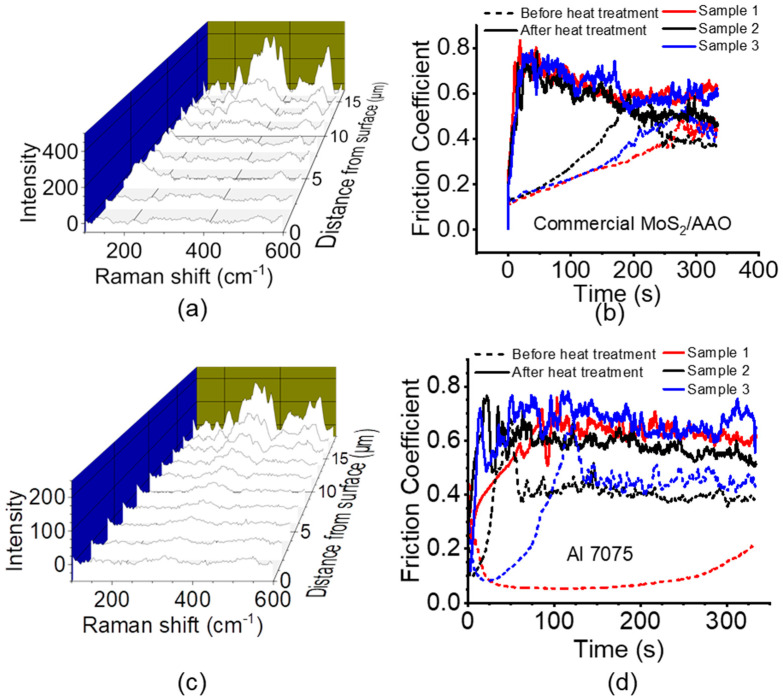
Commercial MoS_2_/AAO comparison. (**a**,**c**) Raman spectroscopic analysis of (**a**) the commercial MoS_2_/AAO film and (**c**) 120-min high-acid anodized Al 7075 MoS_2_/AAO film with no acid pretreatment before heat treatment. (**b**,**d**) Reciprocating scratch tests on commercial MoS_2_/AAO (**b**) with three scratches performed and 120-min high-acid anodized Al 7075 MoS_2_/AAO films before and after heat treatment (**d**) with one scratch per sample shown.

**Figure 10 nanomaterials-14-00451-f010:**
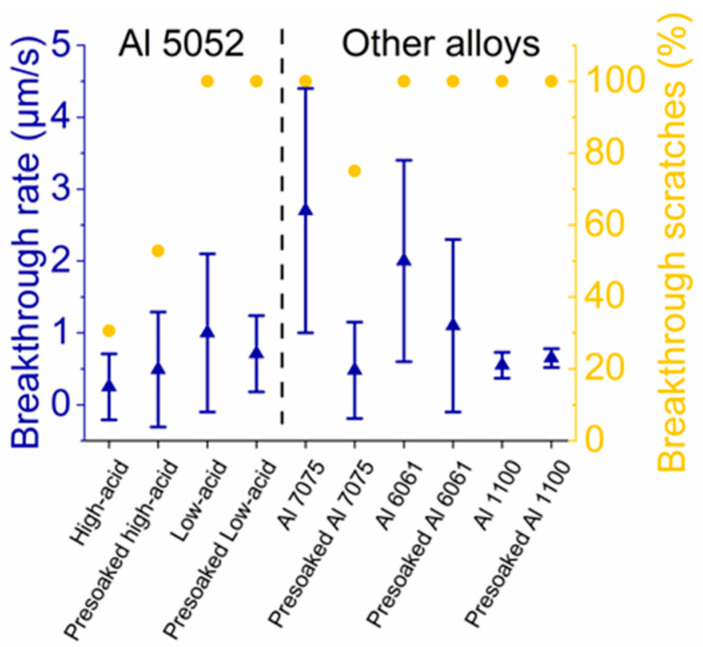
Reciprocating scratch results of MoS_2_ films from different alloys and treatment procedures.

**Figure 11 nanomaterials-14-00451-f011:**
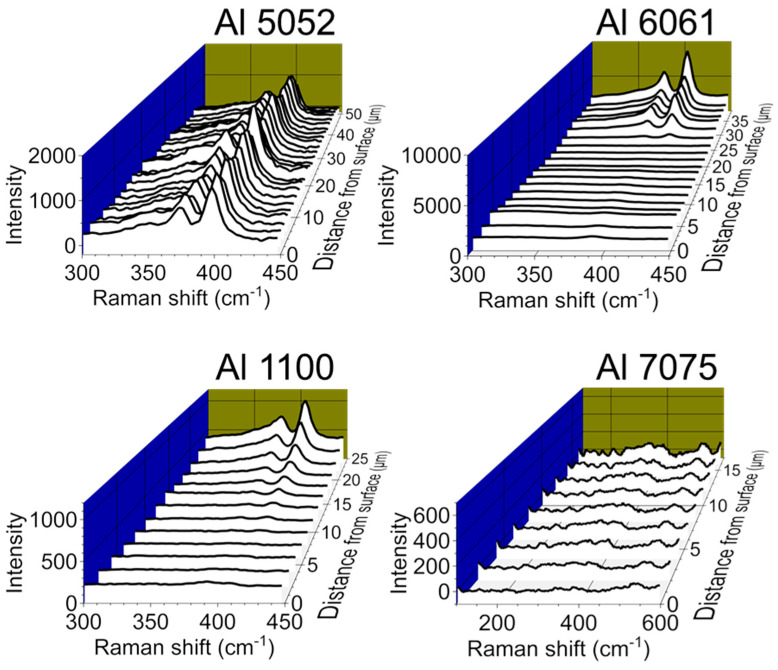
Cross-sectional Raman spectroscopic comparison of MoS_2_/AAO prepared with different aluminum alloys. All samples were 120-min high-acid anodized with no HNO_3_ pretreatment.

**Table 1 nanomaterials-14-00451-t001:** Film thicknesses for different sample preparation parameters (high-acid anodization).

Sample Treatment	AAO/MoS_2_ Thickness (mm)
No pretreatment, 20 min. anodization	14 ± 4
HNO_3_ presoak, 20 min. anodization	15 ± 2
No pretreatment, 120 min. anodization	38 ± 12
HNO_3_ presoak, 120 min. anodization	47 ± 18

**Table 2 nanomaterials-14-00451-t002:** Tribological results for high-acid anodized 5052 Al.

Sample Identification	Breakthrough Rate (mm/s)	Breakthrough Scratches (%)
No pretreatment, 20 min. anodization,	0.28 ± 0.20	83
HNO_3_ presoak, 20 min. anodization	0.22 ± 0.19	63
No pretreatment, 120 min. anodization,	0.25 ± 0.46	31
HNO_3_ presoak, 120 min. anodization	0.49 ± 0.8	53
No MoS_2_, 120 min. anodization	0.12 ± 0.02	100
Heat treated, No MoS_2_, 120 min. anodization	0.7 ± 0.5	100

**Table 3 nanomaterials-14-00451-t003:** Elemental content and friction coefficients of studied aluminum alloys.

Alloy	Al	Mg	Si	Cu	Fe	Cr	Zn	Friction Coefficient
5052	97.2	2.5	-	-	-	0.25	-	0.73 ± 0.03
7075	90	2.5	-	1.6	-	0.23	5.6	0.43 ± 0.01
6061	97.9	1.0	0.6	0.28	-	0.2	-	0.59 ± 0.01
1100	99.2	-	0.2	0.1	0.5	-	-	0.95 ± 0.01

## Data Availability

The original contributions presented in the study are included in the article/Appendix A, further inquiries can be directed to the corresponding author.

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
