# Peer review of "Correlation of Fabrication Methods and Enhanced Wear Performance in Nanoporous Anodic Aluminum Oxide with Incorporated Molybdenum Disulfide (MoS2) Nanomaterials"

_nanomaterials, 2024, doi:10.3390/nano14050451_

Round 1

Reviewer 1 Report

Comments and Suggestions for Authors

(1)   Based on the results of Figure 3, the friction time was only 5 minutes and some COF curves were not stable. So, it is recommended to supplement long-term experimental data to improve the experiment.

(2)   Figure 4 should be modified to clearly show the experimental data.

(3)   In order to better reflect the latest theories and achievements, the relevant research in this area in recent years should be added.

(4)   The conclusion can add the impact and significance of this research work, focusing on the innovation points and important findings.

Comments on the Quality of English Language

Minor editing of English language is required.

Reviewer 2 Report

Comments and Suggestions for Authors

Authors investigated the treatment parameters and Al alloy content in MoS2/AAO films and gives fundamental insight on deposition mechanisms and film structure. It could guide future development of nanoporous materials for various applications. However, the current form of this study cannot be acceptable. Some aspects as listed below:

1: Figure S1 (b) is of low clarity, and it is hoped that it will be replaced with a high-resolution image.

2: The curves in Figure 2 have similar trends, but there are still large differences, and whether this has an impact on the results of the other tests.

3: Is there a correlation between the differences in COF between samples in the same process in Fig. 3 and the V-t curves in Fig. 2?

4: Figure 7 is almost exactly the same as Figure 3, please double check that this is not a mistake.

5: It is necessary to analyze how heat treatment affects film properties.

Round 2

Reviewer 2 Report

Comments and Suggestions for Authors

Authors have revised the manuscript according to the reviewers’ comments. The contents and discussions have been improved significantly.